# Effect of Zoapatle (*Montanoa tomentosa*) on Inflammatory Markers in a Murine Model of Ventricular Hypertrophy

Carlos Enrique López-Luna [1], Cruz Vargas-De-León [1,2], Rocio Alejandra Gutiérrez-Rojas [3], Karla Aidee Aguayo-Cerón [1], Claudia Camelia Calzada-Mendoza [1], Fengyang Huang [4], Rodrigo Romero-Nava [1,*] and Maria Esther Ocharan-Hernandez [1,*]

[1] Sección de Estudios de Posgrado e Investigación, Escuela Superior de Medicina, Instituto Politécnico Nacional, Mexico City 11340, Mexico; jimmyubermensch@live.com (C.E.L.-L.); leoncruz82@yahoo.com.mx (C.V.-D.-L.); aidee.aguayo@gmail.com (K.A.A.-C.); cccalzadam@yahoo.com.mx (C.C.C.-M.)

[2] División de Investigación, Hospital Juárez de Mexico, Mexico City 07760, Mexico

[3] Escuela Nacional de Ciencias Biológicas del Instituto Politécnico Nacional, Mexico City 07738, Mexico; ross.grojas.22@gmail.com

[4] Laboratorio de Investigación en Obesidad y Asma, Hospital Infantil de México Federico Gómez, Mexico City 06720, Mexico; huangfengyang@gmail.com

[*] Correspondence: rromeron@ipn.mx (R.R.-N.); estherocharan@gmail.com (M.E.O.-H.)

**Abstract:** Zoapatle, a native plant utilized for centuries in traditional Mexican medicine, is abundantly found in Mesoamerica and northern South America. Pleiotropic effects of this genus have been recognized, primarily inducing alterations in smooth muscle contractility in animal models. The aim of this study was to evaluate the effect of Zoapatle on the hypertrophy index and the gene expression of TNF-$\alpha$, IL-1$\beta$, NF-$\kappa$B, STAT5, and the PRLR in the brain, left ventricle, and renal cortex of rats with isoproterenol-induced cardiac hypertrophy. Three groups were studied, the control group ($n = 4$), hypertrophy group ($n = 4$) and hypertrophy group treated with Zoapatle ($n = 4$). A ventricular hypertrophy model was developed with 150 mg/kg/day of isoproterenol intraperitoneally administered over two days with a 24 h interval between applications. Zoapatle was administered for 28 consecutive days (25 mg/kg). Gene expression was determined with RT-qPCR. Subsequently, a principal component analysis (PCA) was performed using the RNA expression variables. A notably reduced left ventricle mass index was observed in the Zoapatle group. Additionally, Zoapatle administration in cardiac hypertrophy demonstrated a significant decrease in the gene expression of TNF-$\alpha$, IL-1B, STAT 5, and the PRLR. TNF-$\alpha$ and the transcription factor STAT5 exhibited a similar trend in both the left ventricle and renal cortex, suggesting a correlation with the inflammatory state in these tissues due to ventricular hypertrophy. The findings suggest that Zoapatle reverses the hypertrophy index in a hypertrophy model, concurrently reducing several proinflammatory mediators associated with the hypertrophy index.

**Keywords:** Zoapatle; cardiac hypertrophy; TNF-$\alpha$; NF-$\kappa$B; PRLR

## 1. Introduction

Cihuapatli, a word derived from the Mexica language that means "women's medicine" in English [1], also called the Mexican Zoapatle, has been used for centuries in traditional Mexican medicine, for the induction of labor by increasing uterine activity in terms of tone and frequency, as well as to lessen the effects of dysmenorrhea [2–4]. The genus Montanoa is one of the most prestigious genera of Asteraceae from the mountains of Mesoamerica and northern South America; it is widely distributed in Mexico. The Tomentosa species has three subspecies: xanthiifolia, microcephala, and tomentosa [5]. The pleiotropic effects of the Montanoa genus have been identified, among which are contraceptive effects [2,3], embryotoxicity [6,7], cervical dilation [2,8], increased uterine contractility [2], and the control of uterine bleeding [9]. *Montanoa tomentosa* has opposite effects in different animal

models, for example, in vitro inhibition, a decrease in spontaneous contractions of rat uterine tissue [2], and an increase in uterine contractility in guineapigs and cats [10]. Recent investigations have indicated the presence of aphrodisiac properties in male rats, leading to heightened motivation and enhanced performance in sexual activity [1,8,11]. In addition, beneficial anxiolytic-like actions have been showed with the participation of the subunit GABA-A receptor, which was demonstrated in models to be entirely blocked by picrotoxin and bicuculine [12,13]. It has been reported that the effect of Zoapatle can modulate mood states [7,14] and also exerts a pro-ejaculatory effect acting directly on the spinal system through an oxytocin-like effect [15]. It has been shown that Zoapatle at low doses (3.0 mg/kg) generates anxiolytic-like actions, and at higher doses, sedative effects (>12 mg/kg) have been observed; the likely signaling mechanism is carried out through GABA-A receptors [16].

More than 70 compounds of this genus have been characterized [17]. The most relevant compounds of Montanoa are terpenoids, diterpenoids, monoterpenoids, and sesquiterpenoids [18–20] with anxiolytic [13] antitumor, anti-inflammatory, and antineoplastic properties [21]. It has been described that Montanoa contains triterpenoids with anti-tuberculosis [22], anti-fungal, anti-inflammatory, and anti-viral activity [21]. Three tetracyclic diterpenes (kaurenoic, kauradienoic, and monoginoic acid) have been isolated with different activities, among which are uterotonic [22,23], antispasmodic [24], antineoplastic [25], and antibacterial/antifungal effects [26], and anti-inflammatory activity [27].

The exact mechanism by which Montanoa exerts its effect has not been completely specified. *Montanoa tomentosa* exhibits two distinct in vitro activities: a uterotonic effect and the inhibition of spontaneous contractions in guineapig uterine muscle [28,29]. The compounds in Zoapatle exert their impact on uterine muscle through the direct stimulation of β-adrenergic receptors and/or cholinergic receptors in the smooth muscle cell membrane [29]. This modulation is particularly relevant in cardiovascular diseases, in which the adrenergic pathway plays a hegemonic role in physiological cardiac function, mainly in the β1 receptor [30]. Nevertheless, the exact receptor of this pathway in which *Montanoa tomentosa* acts is still unknown. In contrast, kaurane-type diterpenes, one of the multiple compounds of this genus, inhibit the contractility and induce the relaxation of smooth muscle [1]. Notably, certain components of the Montanoa genus, including Zoapatle, may contribute to inhibitory activity against NF-kappa B, a key player in the inflammatory process [27].

In Mexico, chronic degenerative diseases are predominant among the adult population, including cardiovascular diseases, which represent the most common cause of mortality [29]. One of the main compensation mechanisms of the heart in the face of work-overload stimuli is cardiac hypertrophy, which is a physiological adaptation of the myocardial tissue that is characterized by the elongation of the muscle fibers, improving the coupling, and contraction to preserve the cardiac structural architecture [30–33]. In pathologies such as obesity, cardiac output and blood volume increase in the same way as the pressure and end-diastolic volume of the left ventricle, generating an adaptation of the myocardium by increasing the contractile elements in the myocardial mass (hypertrophy) [31].

Pathological hypertrophy is characterized by cardiomyocyte death and compensatory fibrotic remodeling, which generates a diastolic dysfunction that can progress to heart failure [34]. Cardiac hypertrophy can be classified into three variants: symmetric, eccentric, and concentric, which are related to the geometric structure of the heart [35]. Hypertrophy can be characterized by a disproportionate increase in muscle length, which develops due to ventricular overload and shows an increase in ventricular volume with an associated growth of the interventricular wall and septum (eccentric) [36], or by an increase in the wall thickness, which occurs in hypertension (concentric) [37]. Multiple studies have evaluated the inflammatory process as a therapeutic target to reduce or prevent myocardial hypertrophy in heart failure [38,39]. Inflammation plays a key role in ventricular hypertrophy, but the possible cellular mechanisms involved have not been entirely elucidated. The partici-

pation of many pro-inflammatory cytokines such as IL-6, IL-1β, IL-1RA, and TNF-α has been demonstrated, as has the activation of the NF-κB signaling pathway in hypertrophic cardiac tissue [38]. Reactive oxygen species, vasoactive amines, and eicosanoids play a determining role in ventricular hypertrophy in animal models [13,40,41]. Currently, the exploration of substances that have an anti-inflammatory potential on the progression of ventricular hypertrophy continues [40,42].

The objective of this study was to assess the impact of Montanoa on the expression of genes involved in the inflammatory process, including nuclear factor κβ (NF-κβ), tumor necrosis factor (TNF), interleukin 1β (IL-1β), activator of transcription 5 (STAT5), and the prolactin receptor (PRLR) in an animal model of ventricular cardiac hypertrophy. Additionally, the investigation aimed to correlate these findings with the clinical expression concerning the hypertrophy index and identify the primary inflammatory components associated with ventricular hypertrophy development.

## 2. Materials and Methods

### 2.1. Ethical Approval

All the procedures for the use of laboratory animals described in this study were approved by the Postgraduate Bioethics Committee of our institution (ESM.CI03/07-27-2015) and were carried out in accordance with the regulations of the Mexican Official Standard (41). Male Wistar rats of 200 to 250 g body weight were obtained from CINVESTAV Coapa (Mexico City, Mexico). The animals were kept under standard conditions in a 12 h light/dark cycle, with food and water ad libitum.

### 2.2. Experimental Design

The induction of cardiac hypertrophy in rats was conducted based on the method of the Federal University of Ceará (Laboratory of Experimental Surgery) [40]. We used three groups of animals: the control group ($n = 4$), hypertrophy group (isoproterenol, $n = 4$) and hypertrophy group treated with Zoapatle (isoproterenol + Zoapatle, $n = 4$). The control group was administered with a saline solution of 2 mL intraperitoneally for two consecutive days, each 24 h. The hypertrophy model was initiated by the administration of 150 mg/kg/day of isoproterenol intraperitoneally for two days with a 24 h interval between applications ($n = 8$). The hypertrophy group was divided into two groups, a group without treatment ($n = 4$) and a group treated with Zoapatle ($n = 4$). The hypertrophy group treated with Zoapatle was administered with an aqueous extract of *Montanoa tomentosa*, which was collected from its natural habitat in the state of Tlaxcala, Mexico. Zoapatle leaves were dried for 20 days, then 100 g of material was sprayed in a 1 L of solution and mixed with 1 L of distilled water and heated for approximately 10 min. The infusion was filtered and oven-dried at 55 °C, and the residue produced from the extract was calculated at 80 or 85 mg. Treatment was administered once daily for a total of 28 days with 1 mL/kg (25 mg/kg) through a cannula [42]. The heart was removed, weighed, and then cardiac dimensions were measured. For each animal, the cardiac index was calculated by dividing the weight of the heart by the body weight [43]. Also, samples of brain tissue, the left ventricle, and the renal cortex were collected.

### 2.3. Total RNA Extraction and cDNA Synthesis

Tissues from the left ventricle, brain, and renal cortex were extracted and frozen in liquid nitrogen. Subsequently, the tissues of each group were lysed by treatment with 500 μL of guanidinium thiocyanate (TRIzol, Invitrogen) according to the manufacturer's instructions. Then, 200 μL of chloroform was added, incubated for 5 min, and centrifuged at 13,000 rpm at 4 °C for 15 min. The aqueous phase was extracted, to which 500 μL of isopropanol was added and incubated for 15 min at −20 °C. Finally, two washes were carried out with 70% ethanol. Total RNA was suspended with 30 μL of PCR water. The concentration and purity of the total RNA were quantified using a nanophotometer (Implen, Inc., Estlake Village, CA, USA) at optical densities at 260/280 and 260/230 nm. A purity

ratio of 1.8–2.2 was required for these studies. Electrophoresis was performed, and two bands (18 s and 28 s) were identified to determine the integrity of the RNA in agarose gel. The gel image was visualized and digitized using an E-gel Imager (Invitrogen, Carlsbad, CA, USA). Reverse transcription (RT) was catalyzed with M-MLV Reverse Transcriptase (Invitrogen, Carlsbad, CA, USA), using 500 ng of total RNA in a total volume of 20 μL. The reaction mixture was incubated in a thermal cycler (Techne, Staffordshire, UK) according to the manufacturer's instructions; cDNA was used for the expression analysis of genes of interest.

### 2.4. RT-qPCR

The reference and interest genes are listed in Table 1. The primers for mRNA were designed in the Universal Probe Library Assay Design Center (LifeScience, Roche, Germany) and were synthesized by Sigma Aldrich. NCBI Blast was used to check the specificity of the primers. RT-qPCR analyses were performed on a Prime Pro 48 Real-Time PCR apparatus (TECHNE) using FastStar SYBR Green Master (Roche) for detection in a final volume of 10 μL. Then, 5 μL of 5x reaction buffer was added to the reaction mixture, as was 0.3 μL of both the forward and reverse primers (Table 1) along with 0.15 μL of SYBR Green, and the remainder was filled with water to a final volume of 9 μL. Subsequently, 1 μL (1000 ng of reverse transcription RNA) of cDNA was added. PCR reactions were initiated with a denaturation step of 10 min at 95 °C, followed by 45 cycles of amplification (denaturation for 10 s at 95 °C, hybridization for 40 s at 60 °C, and extension for 10 s at 72 °C). A dissociation protocol with a gradient (0.5 °C every 30 s) from 65 °C to 95 °C was used to investigate the specificity of the reaction and the presence of dimers between primers, and the specific amplification of the gene was confirmed with a single peak in the melt curve analysis. The mRNA expression levels of the genes were determined using the $2^{-\Delta\Delta Ct}$ comparative method [44], normalizing the expression with β-actin and HPRT.

**Table 1.** Primer sequences used for quantitative RT-qPCR.

| Genes | Primer Sense | Primer Antisense | GenBank |
|---|---|---|---|
| STAT5a | 5′-AATGAACAGAGGCTGGTCCG-3′ | 5′-GCAGCTCCTCAAACGTTTGG-3′ | NM_017064.2 |
| TNF-α | 5′-TGAACTTCGGGGTGATCG-3′ | 5′-GGGCTTGTCACTCGAGTTTT-3′ | NM_012675.3 |
| IL-1b | 5′-CCCCAAAAGATTAAGGATTGC-3′ | 5′-AGCTGGATGCTCTCATCTGG-3′ | NM_031512.2 |
| PRLR | 5′-CAGTTCCTGGGCCAAAAATA-3′ | 5′-CAAGGCACTCAGCAGCTCTT-3′ | NM_001034111.1 |
| NF-kb1 | 5′-ACAGCTGGATGTGTGACTGG-3′ | 5′-TCCTTCCCAGACTCCACCAT-3 | NM_001276711.1 |
| B-Actin | 5′-CCCGCGAGTACAACCTTCT-3′ | 5′-CGTCATCCATGGCGAACT-3′ | NM_031144.3 |

STAT5, signal transducer and activator; TNF-α, Tumor necrosis factor alpha; IL-1b, Interleukin 1 beta; PRLR, prolactin receptor; NF-kB, factor kappa B; B-Actin (Housekeeping).

### 2.5. Principal Component Analysis (PCA) of Genes in Hypertrophy

PCA is a method of dimension reduction that is used to find clusters based on gene expression and identify which of the genes contributes the most to clustering [45,46]. Subsequently, quantify the strength of the association between ventricular mass index and principal component scores.

### 2.6. Statistical Analysis

Data are presented as mean ± SEM. The statistical analysis was performed using GraphPad Prism Version 7.0 (GraphPad Software, Inc., La Jolla, CA, USA). Inter-group differences were analyzed with a one-way analysis of variance, followed by Tukey's post-hoc test. $p < 0.05$ was considered to indicate a statistically significant difference. Subsequently, a PCA was conducted using the RNA expression variables. The PCA was conducted using R Statistical Software version 4.1 with the R package FactoMineR [47], factoextra [48], and ggplot2 [49]. We used the Kaiser–Meyer–Olkin (KMO) measure of sampling adequacy for the PCA; KMO values < 0.6 indicated inadequate sampling. Finally, Pearson's linear

correlation was used to measure the association between the ventricular mass index and principal component scores.

## 3. Results

### 3.1. Isoproterenol-Induced Cardiac Hypertrophy

We determined the mass of the left ventricle index to evaluate cardiovascular hypertrophy and the utility of the model; we observed a significant increase in the index value in the group treated with ISO compared with that in the control group. When we evaluated the effect of Zoapatle in cardiovascular hypertrophy by assessing the left ventricle mass index, we observed that the group that received Zoapatle showed a significantly lower index compared with the group treated with ISO and even the control group (Figure 1).

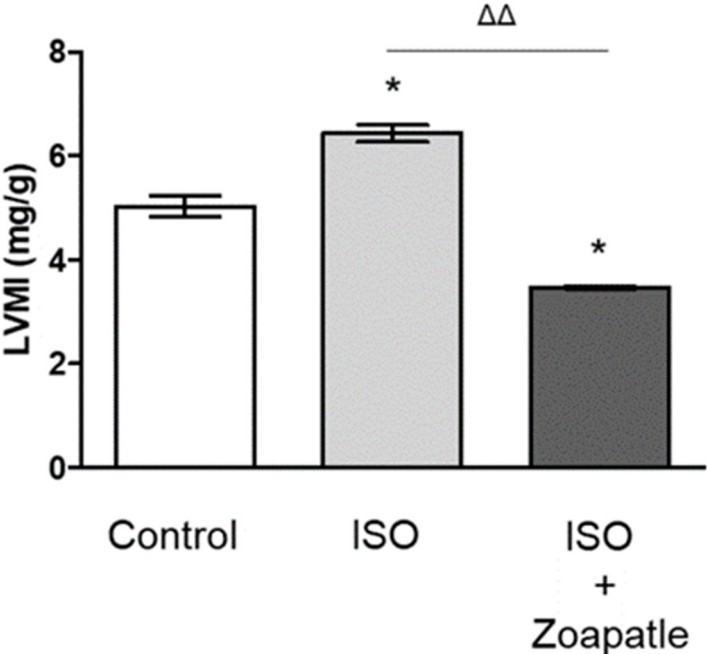

**Figure 1.** LMIM, left ventricular mass index. Control group; ISO (hypertrophy by isoproterenol group); ISO + Zoapatle (hypertrophy by isoproterenol + Zoapatle group). Lines represent the ventricular mass index in each group; median $\pm$ standard error. * $p < 0.05$ versus without hypertrophy (control group). $^{\Delta\Delta}$ $p < 0.01$ versus hypertrophy (ISO group).

### 3.2. Gene Expression in the Left Ventricle

To the investigate the potential role of interleukins (TNF-$\alpha$ and IL-1$\beta$) (Figure 2a,b), transcription factors (NF-$\kappa$B and STAT5) (Figure 2c,d), and the prolactin receptor (PRLR) (Figure 2e) in cardiac hypertrophy, we first explored the changes in gene expression in this pathology and then how the use of Zoapatle modulated that expression. We found the administration of ISO increased the expression of TNF-$\alpha$, IL-1B, STAT 5, and the PRLR in the left ventricle compared with that in the control group (without isoproterenol), while the expression of NF-$\kappa$B decreased. On the other hand, the use of Zoapatle in this model of cardiac hypertrophy decreased the expression of TNF-$\alpha$, IL-1B, STAT 5, and the PRLR, but increased the expression of NF-$\kappa$B when compared with that in the hypertrophic group (ISO).

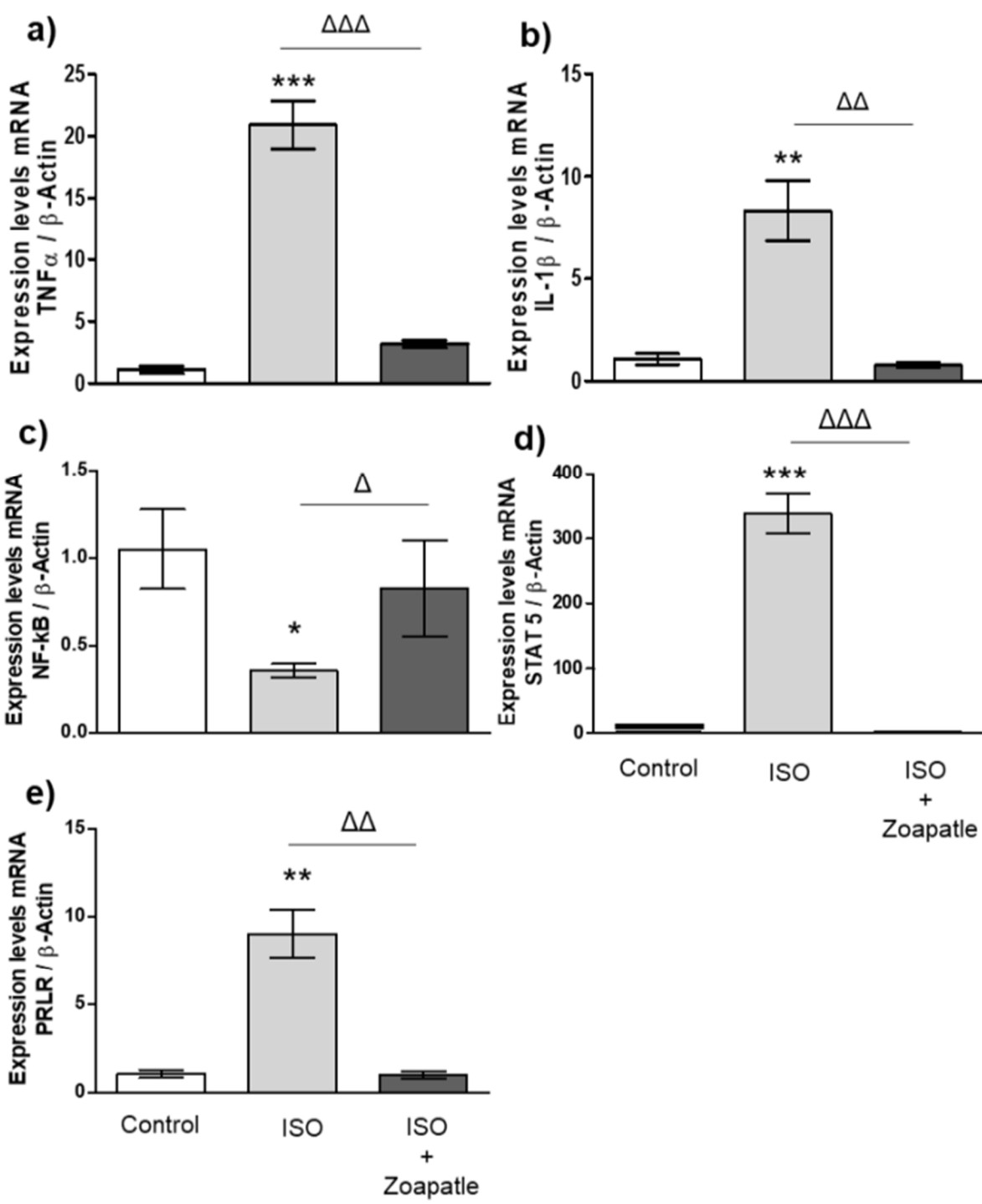

**Figure 2.** Gene expression in the left ventricle. Control group; ISO (hypertrophy by isoproterenol group); ISO + Zoapatle (hypertrophy by isoproterenol + Zoapatle group). (**a**) TNF-α, tumor necrosis factor alpha; (**b**) IL-1b, Interleukin 1-beta; (**c**) NF-kB, nuclear factor kappa B; (**d**) STAT5, signal transducer and activator; (**e**) PRLR, prolactin receptor; B; B-Actin. Lines represent median ± standard error. mRNA levels were normalized to B-actin. * $p < 0.05$, ** $p < 0.01$, *** $p < 0.001$ versus control. $^{\Delta}$ $p < 0.05$, $^{\Delta\Delta}$ $p < 0.01$, $^{\Delta\Delta\Delta}$ $p < 0.001$ versus hypertrophy (ISO group).

### 3.3. Gene Expression in the Brain

We observed significant increases in the messenger RNA of TNF-α (Figure 3a) and the PRLR (Figure 3e) in left ventricular hypertrophy induced by ISO when compared with that of the control group. Furthermore, the expression of IL-1β (Figure 3b) decreased,

while NF-κB (Figure 3c) and STAT5 (Figure 3d) expressions did not change compared with those of the non-hypertrophic group. On the other hand, treatment with Zoapatle led to a significant increase in the expression of TNF-α and IL-1β compared with that of the ISO group. However, we observed that the administration of Zoapatle decreased PRLR expression compared with that of the hypertrophic group, while the expression of NF-κB and STAT 5 did not change.

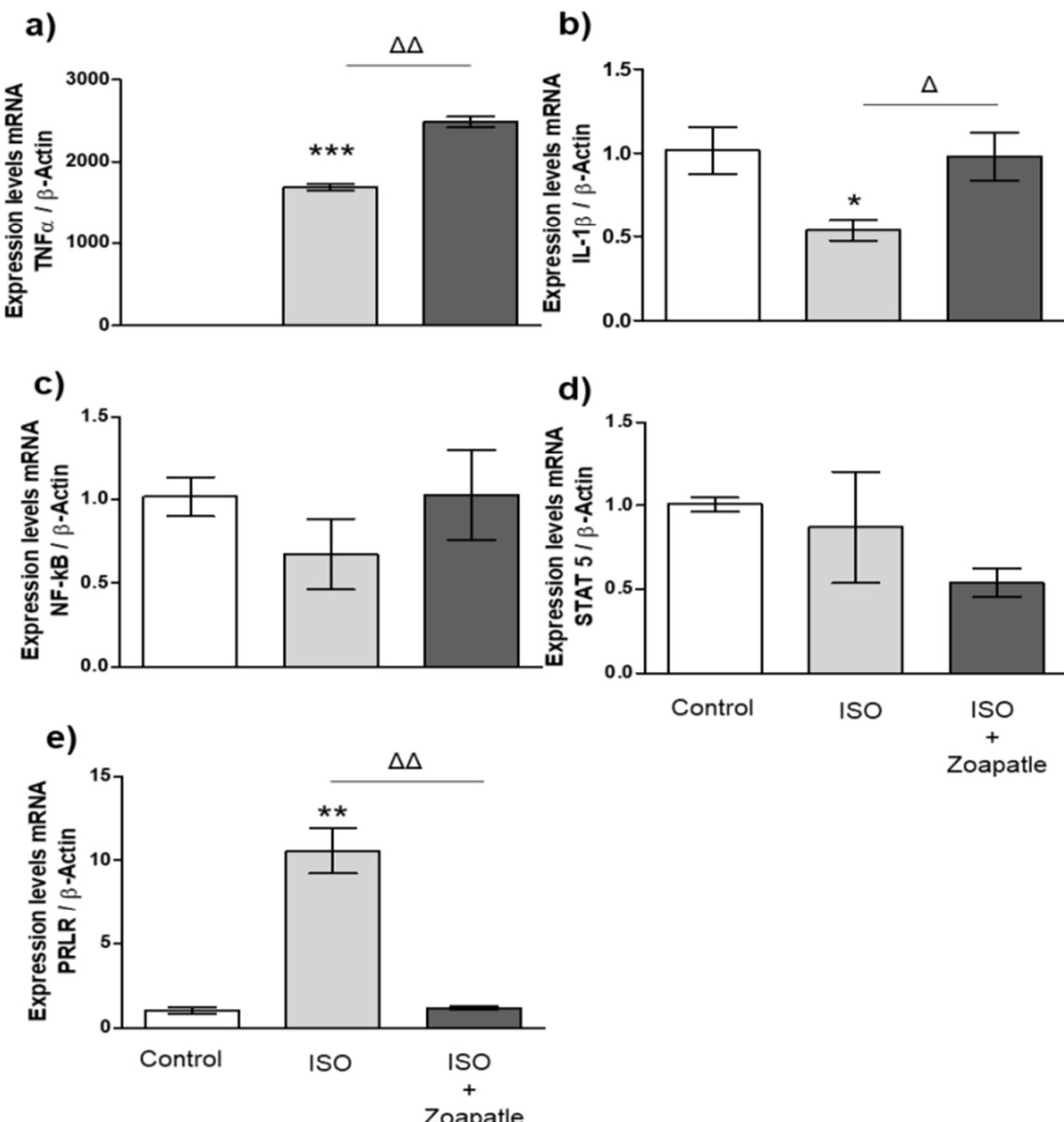

**Figure 3.** Gene expression in brain. Control group; ISO (hypertrophy by isoproterenol group); ISO + Zoapatle (hypertrophy by isoproterenol + Zoapatle group). (**a**) TNF-α, tumor necrosis factor alpha; (**b**) IL-1b, Interleukin 1-beta; (**c**) NF-kB, nuclear factor kappa B; (**d**) STAT5, signal transducer and activator; (**e**) PRLR, prolactin receptor; B; B-Actin. Lines represent median $\pm$ standard error. mRNA levels were normalized to B-actin. * $p < 0.05$, ** $p < 0.01$, *** $p < 0.001$ versus control. $^{\Delta}$ $p < 0.05$, $^{\Delta\Delta}$ $p < 0.01$ versus hypertrophy (ISO group).

### 3.4. Gene Expression in the Renal Cortex

The administration of isoproterenol increased the expressions of TNF-α (Figure 4a), NF-κB (Figure 4c), STAT5 (Figure 4d), and the PRLR (Figure 4e) compared with those of the control group, while IL-1β expressions (Figure 4b) did not show changes. On the

other hand, treatment with Zoapatle significantly reduced the expressions of TNF-α and STAT5 messenger RNA compared with those of the group that received ISO; however, we observed a significant increase in the expression of NF-κB.

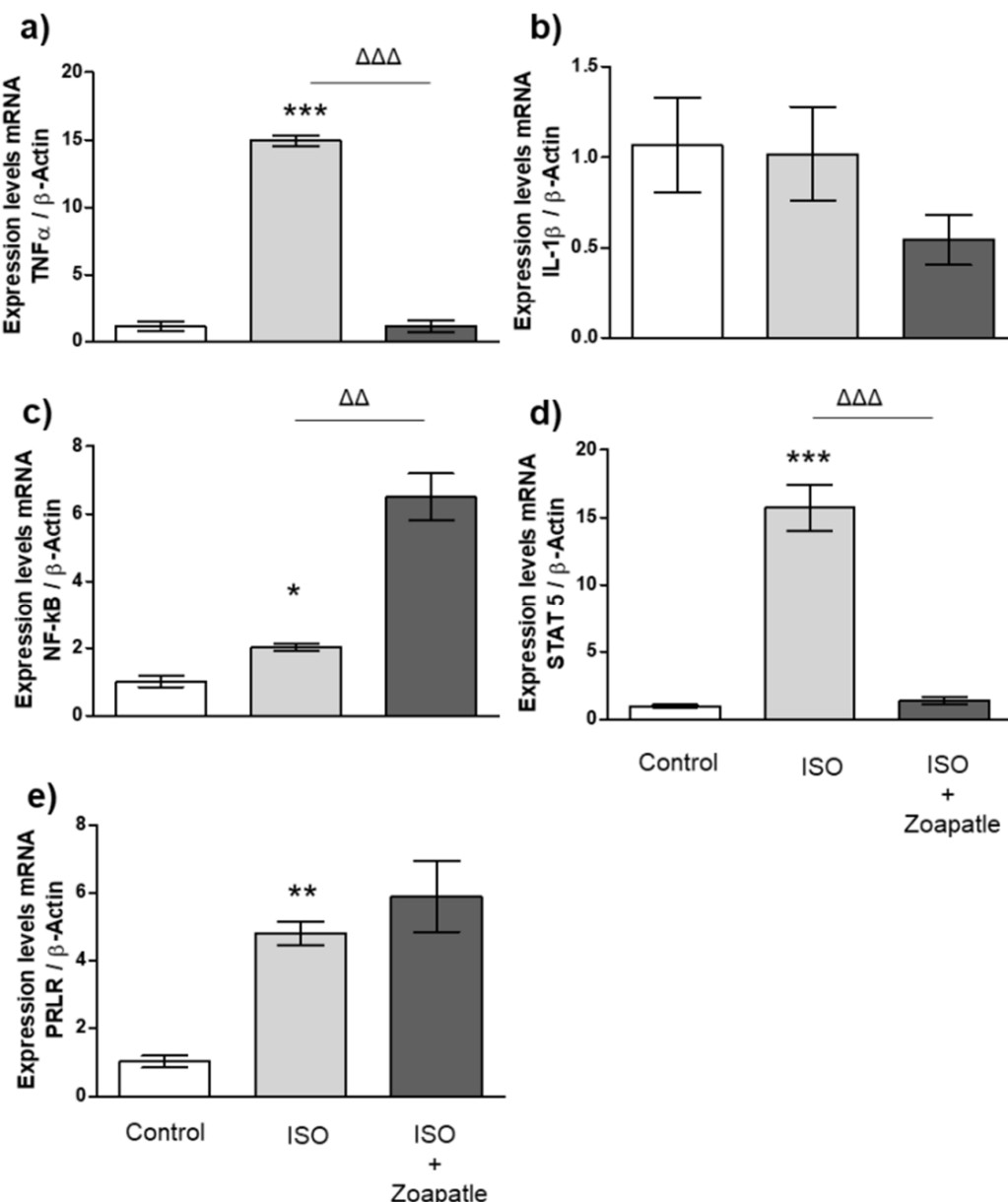

**Figure 4.** Gene expression in renal cortex. Control group; ISO (hypertrophy by isoproterenol group); ISO + Zoapatle (hypertrophy by isoproterenol + Zoapatle group). (**a**) TNF-α, Tumor necrosis factor alpha; (**b**) IL-1b, Interleukin 1-beta; (**c**) NF-kB, nuclear factor Kappa B; (**d**) STAT5, Signal transducer and activator; (**e**) PRLR, Prolactin receptor; B; B-Actin. Lines represent median ± standard error. mRNA levels were normalized to B-actin. * $p < 0.05$, ** $p < 0.01$, *** $p < 0.001$ versus control. ΔΔ $p < 0.01$, ΔΔΔ $p < 0.001$ versus hypertrophy (ISO group).

*3.5. Principal Component Analysis*

The Kaiser–Meyer–Olkin value is suitable for PCA, 0.7. Subsequently, a PCA was implemented to perform a dimensional reduction of five RNA expression variables. The first two principal components explained 97.2% of the variability. Table 2 shows the variable loading and correlation coefficients for the principal component scores. These components were integrated as follows. First component: STAT5, IL1b, the PRLR, and TNF; second component: NF-κB. Figure 5a shows that the first two principal components formed two

clusters; one cluster was formed by the subjects in the ISO group and a second cluster was formed by the control and ISO + Zoapatle groups. The first principal component had a strong correlation with the ventricular mass index (LVMI): 0.79 (95% CI: 0.27–0.95, $p = 0.011$), as shown in Figure 5b.

**Table 2.** PCA on the RNA expressions.

| Genes | First Component Loading (Correlation) | Second Component Loading (Correlation) |
|---|---|---|
| STAT5 | 0.477 (0.999 **) | 0.126 (0.093) |
| TNF | 0.466 (0.968 **) | 0.103 (0.076) |
| IL-1b | 0.466 (0.967 **) | 0.266 (0.197) |
| PRPL | 0.465 (0.966 **) | 0.197 (0.146) |
| NF-kb | −0.348 (−0.710 *) | 0.929 (0.689 *) |

Results from PCA on the RNA expressions. For the first principal component, all variables had a correlation greater than 0.95 (**), except for NF-kB (*). * $p < 0.05$, ** $p < 0.01$ versus control. For the second component, only NF-kB had the highest correlation.

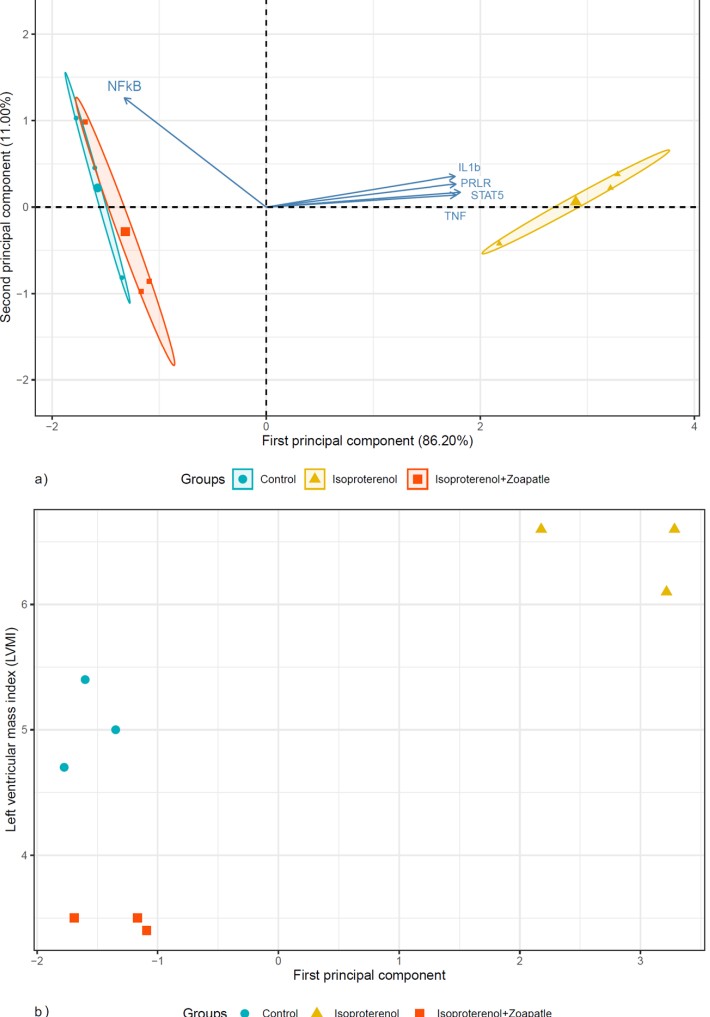

**Figure 5.** The biplot shows the distribution of the subjects according to the two first principal components. (**a**) The circle, triangle, and square are subjects in the control, isoproterenol, and isoproterenol + Zoapatle groups. (**b**) The scatter plot between the first principal component and the left ventricular mass index.

## 4. Discussion

Cardiac hypertrophy is a common morphologic expression of the development of an ongoing cardiovascular disease such as heart failure [50–54]. Currently, there are no medications that specifically prevent the development of cardiac hypertrophy, but there are medications to manage the underlying pathology, like systemic arterial hypertension, some of which have been shown to reduce cardiac remodeling typical of the natural history of the disease, among which are angiotensin-converting enzyme inhibitors, angiotensin II receptor blockers [52], and angiotensin receptor-neprilysin inhibitors, and in patients with established heart failure, the use of diuretics like sodium–glucose transporter 2 inhibitors is helpful to reduce symptomatology [54]. The present study evaluated the effect of Zoapatle on the expression of TNF-$\alpha$, IL-1$\beta$, NF-$\kappa$B, STAT5, and the PRLR in the left ventricle, brain, and renal cortex of rats with ISO-induced ventricular hypertrophy [55]. Our results showed that ISO-induced ventricular hypertrophy increased the gene expression of TNF-$\alpha$, IL-1$\beta$, STAT5, and the PRLR in the left ventricle, while the gene expression of NF-$\kappa$B was decreased compared with that in the control group. TNF-$\alpha$, IL-1$\beta$, and IL-6 (proinflammatory cytokines) have been described to play relevant roles in cardiovascular diseases [56], and it has also described that the PRLR is increased in cardiac hypertrophy [57]. On the other hand, we observed significantly decreased gene expressions of TNF-$\alpha$, IL-1B, STAT5, and the PRLR in the left ventricle of rats with ventricular hypertrophy that received treatment with Zoapatle compared with those in the group with ventricular hypertrophy and almost reached the control group levels. Moreover, NF-$\kappa$B gene expression was significantly decreased compared with that in the hypertrophic group and the control group but reached similar levels to those in the control group with Zoapatle treatment.

In the brain, we observed that the process of ventricular hypertrophy induced by ISO increased the expression of TNF-$\alpha$ and the PRLR, while the expression of IL-1$\beta$ was decreased and showed no changes in the expression of NF-$\kappa$B and STAT5 when comparing the expression levels with those in the control group. However, we observed that the administration of Zoapatle changed the expression of IL-1$\beta$ and the PRLR to be like that in the control group. While the expression of TNF-$\alpha$ showed an increase with Zoapatle treatment, NF-$\kappa$B and STAT5 did not show changes. So far, there is no evidence in animal models that Zoapatle modulates any of these genes in the brain during the development of a ventricular hypertrophy process or of this already established pathology. On the other hand, we observed that ISO at the level of the renal cortex increased the expression of TNF-$\alpha$, STAT5, and the PRLR, while the expression of IL-1$\beta$ did not show changes compared with that in the control group. The use of Zoapatle decreased the expression of TNF-$\alpha$ and STAT5 in the renal cortex of animals with ISO-induced ventricular hypertrophy [58]. We observed that the gene expression of TNF-$\alpha$ and the transcription factor STAT5 had similar results in left ventricle and in the renal cortex, according to other investigations that confirmed the participation of this cytokine in tissues that are in an inflammatory state due to ventricular hypertrophy generated by the administration of isoproterenol [59]. Currently, there is evidence of a strong association between cardiac and renal events, as well as progressive functional deterioration of both organs [59,60]. For example, STAT5 activation might play a facilitative role in the occurrence of cardiac hypertrophy, inhibiting STAT5 attenuated Ang-II-induced cardiac dysfunction, cardiomyocyte hypertrophy, and inflammation in vivo and in vitro [61]. Our results showed that ISO promoted an inflammatory process in different organs, where TNF-$\alpha$ plays a role as a target for the treatment of this pathology, because it is modulated by the administration of Zoapatle in models of cardiac hypertrophy. Nevertheless, there were several limitations in our study, like the small size of study groups and the fact that the model of cardiac hypertrophy was based on a model of a rapid onset of hypertrophy. Moreover, although the inflammation pathway plays a hegemonic role, we could not extrapolate this to a human model, in which appearance occurs very slowly, so it is prudent to carry out new trials with different models of hypertrophy that are more related to the pathophysiology of heart failure. We performed a PCA on gene expression to integrally identify relevant genes modulated by the use of Zoapatle in rats with ventricular

hypertrophy. The PCA analysis showed that two clusters formed: one was formed by the control and ISO + Zoapatle groups, and the other cluster was formed only by the ISO (ventricular hypertrophy) group, which showed the reversal of the damage induced by Zoapatle. In addition, the first principal component had a strong correlation with the left ventricular mass index, which suggested that this panel of RNAs, reduced by PCA, could be used as a non-invasive biomarker of the left ventricular mass index.

In summary, the pharmacological effects of Zoapatle include uterotonic, anxiolytic, anti-inflammatory, and antineoplastic properties. Currently, there is no information on the effect of Zoapatle in the regulation of TNF-α, IL-1β, NF-κB, STAT5, and the PRLR in isoproterenol-induced cardiac hypertrophy models, nor the mechanism by which Zoapatle exerts its effect on the transcription of these genes and the proteomics.

The exact biological mechanisms that underlie hypertension-associated heart failure remain unclear; nevertheless, in patients with heart failure with preserved ejection fraction, it has been documented that changes occur in proteins related to mitochondrial metabolism and the cardiac contractile apparatus. Proteomics analyses of the left ventricular tissue showed an upregulation of ketone body transporters and impairment in phosphorylation in proteins like titin, including changes in sarcomeric proteins, mitochondrial-related proteins, and NAD-dependent protein deacetylasesirtuin-3 (SIRT3) [62]. Isoproterenol is a nonselective beta-adrenergic agonist inducing early cardiac hypertrophy and hypercontractility followed by HF with cardiac dilation and ventricular dysfunction secondary to chronic adrenergic overstimulation in rats [40]. In this isoproterenol murine model of cardiac hypertrophy, several proteins are specifically and significantly altered, like a reduction in protein MYL2 and MYL3 and Desmin, and an increase in heat shock proteins like HSP60, HSP70, HSPD1, and prohibitin, which is associated with mitochondrial dysfunction [63]. Further research is needed to elucidate alterations in the proteome profile of isoproterenol-induced hypertrophy and Zoapatle.

Currently, there is a wide range of studies on the subject of cardiac hypertrophy and heart failure. This study showed multiple therapeutic objectives not only to combat the symptoms of the already established disease, as is the case with most current treatments, but to prevent the progression of cardiac hypertrophy before manifesting deleterious clinical presentations for the patient. Our data offer new insights into understanding the beneficial effects of *Montanoa tomentosa* at the cardiac level. Therefore, assessing the profile of both gene and protein expressions, as well as the evaluation of biochemical and physiological parameters in animal models and the integration of these in PCAs associated with the hypertrophy index in multiple tissues, are necessary to elucidate the main signaling pathways and genes involved, which could be potential therapeutic targets.

## 5. Conclusions

In conclusion, it was shown that Zoapatle remarkably reverted the hypertrophy index generated by the administration of isoproterenol. Multiple genes increased their expressions with the cardiac hypertrophy model and remarkably decreased their expressions after treatment with Zoapatle. More research is still necessary; however, the current evidence provides a path forward for the search for new therapeutic objectives, not only for the clinical manifestations but also to reduce the progression of the disease before it is triggered.

**Author Contributions:** Conceptualization, M.E.O.-H. and R.R.-N.; methodology, C.E.L.-L., R.A.G.-R. and K.A.A.-C.; software, C.V.-D.-L.; validation, F.H., R.R.-N. and C.C.C.-M.; formal analysis, C.V.-D.-L.; investigation, C.E.L.-L. and K.A.A.-C.; writing—original draft preparation, C.E.L.-L. and K.A.A.-C.; writing—review and editing, R.R.-N., F.H., R.R.-N. and M.E.O.-H.; funding acquisition, M.E.O.-H., C.V.-D.-L. and R.R.-N. All authors have read and agreed to the published version of the manuscript.

**Funding:** This research was partially supported by the Sección de Estudios de Posgrado e Investigación SIP-IPN 20201084, SIP-IPN 20210474 and SIP-20211050.

**Institutional Review Board Statement:** The animal study was approved by the Institutional Review Board of Escuela Superior de Medicina of Instituto Politecnico Nacional (ESM.CI03).

**Informed Consent Statement:** Not applicable.

**Data Availability Statement:** Data are contained within the article.

**Conflicts of Interest:** The authors declare no conflicts of interest.

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
