# Peer review of "Effect of Zoapatle (Montanoa tomentosa) on Inflammatory Markers in a Murine Model of Ventricular Hypertrophy"

_scipharm, doi:10.3390/scipharm92010009_

Round 1
Reviewer 1 Report
Comments and Suggestions for Authors
To assess the scientific validity of the manuscript under consideration, it is imperative that the authors furnish substantiating proof of the primer pair's specificity for the designated target genes. The primer pairs were subjected to scrutiny using the UCSC In-Silico PCR tool (https://genome.ucsc.edu/cgi-bin/hgPcr), which, notably, yielded no matches for the primers listed in the study. This underscores the necessity for the authors to provide additional evidence of the primers' accuracy and specificity in their experimental design.
Comments on the Quality of English Language-
Reviewer 2 Report
Comments and Suggestions for Authors
Interesting work on the role of inflammatory molecules in the generation of left ventricular hypertrophy, and how a product with anti-inflammatory properties can modulate and even reverse this hypertrophy.
There are several comments I would like to make as a clinical cardiologist (and deeply interested in translational research).
- The abstract could be very improved, the results are poorly expressed and there are sentences in the conclusions that do not stand up from the rest of the abstract. I recommend rewriting completely
- The beginning of the article is incorrect. Cihuapatli is not a Spanish word and in no way does it mean that. It is a word of Mexica origin.
- Page 2, line 88: The biggest problem derived from ventricular hypertrophy is not a “remarkable decrease in the ejection function” but the diastolic dysfunction it produces.
- Page 3, line 103, rewrite the aim of the study correctly.
- Review point 2.2 in full. In no case is it a model of infarction nor are there “infarcted” groups. Correct this important error.
- Line 275: Hypertrophy is not a risk factor; it is an established heart disease per se.
- Discussion section: There should be at least one paragraph on the evidence of the treatment of those pathways mentioned in heart and kidney disease. There should be another paragraph of potential targets and expected outcomes.
- You need a limitations paragraph that comments on the small sample. In this paragraph, it should also be mentioned that the model used could explain the benefit, since it causes rapid onset ventricular hypertrophy, in which inflammation and the pathways used undoubtedly play a role. It should be noted that we do not know if this role can be modulated in the human model, which appears very slowly, and that it should be tested in other models more similar to humans before transferring knowledge.
Reviewer 3 Report
Comments and Suggestions for Authors
The article “The effect of Zoapatle (Montanoa tomentosa) on inflammatory markers in a murine model of ventricular hypertrophy is an interesting topic, however, it is difficult to follow because” it is written in poor English.
For example:
Lines 53-54: Decrease sperm motility does not explain aphrodisiac properties. The paragraph needs to be rephrased.
Line 120: control group without surgery? The authors refer to the control group, I don't understand that they mean by surgery.
Lines 339-341 – “NF-κB is associated with the hypertrophy index in rats treated with Zoapatle as well as rats without hypertrophy” – I don’t understand this conclusion.
Furthermore:
- the introduction should be restructured to reflect more clearly Zoapatle's effects. The data could even be entered in tabular format.
- for the results to reflect the real effect of Zoapatle, the number of animals included in each group must be larger.
The in-text citation of the references and the format of the References are not in accordance with the journal's requirements.
In its current format, the manuscript cannot be accepted for publication.
Round 2
Reviewer 2 Report
Comments and Suggestions for Authors
Changes Have Bing Made and The Paper improve. I Have no More Requests
Author Response
Thank you for your comments.
Reviewer 3 Report
Comments and Suggestions for Authors
In the abstract, the authors say that Zoapatle induces alterations in smooth muscle contractility, which means that it alters contractility, i.e. produces relaxation. Then, in line 50, they state that it increases uterine activity in terms of tone and frequency. In lines 79-80 they say that the effect is dual. Information must be consistent throughout the article.
Lines 62, 63 ”It has been reported that the effect of Zoapatle can modulate mood states (10), acting directly on the spinal system through an oxytocic effect (13), with a similar action to oxytocic drugs”. Instead of oxytocic it should have been written oxytocin. Oxytocic and oxytocin effect are two different things, don't the authors differentiate between the oxytocin effect and the uterine muscle contraction effect (oxytocic)? Furthermore, the oxytocic effect is unrelated to mood, but oxyocin is. Reference 13 refers to the oxytocin and pro-ejaculatory effects.
Line 83 ”This modulation is particularly relevant in cardiovascular diseases”. Does it refer to beta 2 and M1/3 receptor stimulation? These receptors would explain the smooth muscle action referred to in lines 80-81. But, at cardiac level there are beta 1 and M2 receptors. It must be specified which type of connection the authors are referring to? What modulation is relevant in this case?
Lines 285-290 The paragraph needs to be rephrased since there is a major difference between substances that interfere with the action or synthesis of angiotensin 2, WHICH PREVENT THE PROGRESSION of the cardiac hypertrophy, and substances with a diuretic effect, including SLGT2 inhibitors, which improve symptoms.
Lines 327 – hypertrophic animals? Please correct. Hypertrophic animals mean something else.
Lines 341-342 The authors list other effects of Zoapatle than at the beginning of the manuscript. All effects should be specified at the beginning.
Lines 347-351 The entire paragraph should be rephrased.
Comments on the Quality of English LanguageThere are still many spelling and punctuation errors. The manuscript must be reviewed by an English speaker.
Author Response
Reviewer 3 (Round 2)
Question
In the abstract, the authors say that Zoapatle induces alterations in smooth muscle contractility, which means that it alters contractility, i.e. produces relaxation. Then, in line 50, they state that it increases uterine activity in terms of tone and frequency. In lines 79-80 they say that the effect is dual. Information must be consistent throughout the article.
Answer
The effects of Zoapatle in different cells have not been described totally and the studies about that suggest that some of the effects of the plant could be tissue-specific (it means that the effect depends on the site) and the possibility of these effects take place through different pharmacological targets. We have found opposite effects of the plant but in different species and in vitro models and we describe that in the text.
Question
Lines 62, 63 ”It has been reported that the effect of Zoapatle can modulate mood states (10), acting directly on the spinal system through an oxytocic effect (13), with a similar action to oxytocic drugs”. Instead of oxytocic it should have been written oxytocin. Oxytocic and oxytocin effect are two different things, don't the authors differentiate between the oxytocin effect and the uterine muscle contraction effect (oxytocic)? Furthermore, the oxytocic effect is unrelated to mood, but oxyocin is. Reference 13 refers to the oxytocin and pro-ejaculatory effects.
Answer
Thank you for your comments the paragraph has been rephrase and changes in the manuscript.
Question
Line 83 ”This modulation is particularly relevant in cardiovascular diseases”. Does it refer to beta 2 and M1/3 receptor stimulation? These receptors would explain the smooth muscle action referred to in lines 80-81. But, at cardiac level there are beta 1 and M2 receptors. It must be specified which type of connection the authors are referring to? What modulation is relevant in this case?
Answer
Thank you for your comment. The work Dr Gallegos published in the Contraception journal, between 1983 and 1985, about Montanona are the only evidence about this effect in uterine muscle. He described that Montanona tomentosa has an effect on uterine muscle through direct stimulation of b-adrenergic nevertheless there is no evidence about which receptors it stimulates and currently there is not much recent evidence about it
Question
Answer
Thank you for the comment. The manuscript has been modifying.
Question
Lines 327 – hypertrophic animals? Please correct. Hypertrophic animals mean something else.
Answer
Thank you for the comment. The manuscript has been modifying.
Question
Lines 341-342 The authors list other effects of Zoapatle than at the beginning of the manuscript. All effects should be specified at the beginning.
Answer
Thank you for the comment. The manuscript has been modifying.
Question
Lines 347-351 The entire paragraph should be rephrased.
Answer
Thank you for the comment. The manuscript has been modifying.
Round 3
Reviewer 3 Report
Comments and Suggestions for Authors
Please see the attached comments.

Comments on the Quality of English LanguageOnce again, I draw attention to the fact that the manuscript requires proofreading to correct typos, formatting, and grammatical errors.
Author Response
We appreciate your comments, which help us improve the quality of the publication.
1.The authors state that: beneficial anxiolytic-like actions have been showed whit the participation of the GABAA receptor (11,12).
Answer
Thank you for your comment, the manuscript has been modify.
- It refers to the agonistic effect on GABA-A receptors? The expression participation of GABAA receptors is too general.
Answer
Thank you for your comment, we have modified the manuscript with the following legend “benefical anxiolytic-like actions have been showed with the participation of the subunit GABA-A receptor, as were demonstrated in models blocking entirely by picrotoxin and bicuculine”.
- Please correct oxytocin effect with oxytocin-like effect.
Answer
Your suggestion has been modify in the manuscript.
- The authors state that the adrenergic effect is important for cardiac function. It's true, but at cardiac level not the beta 2 receptors are important (receptors located in the smooth muscles), but the beta-1 receptors. I do not understand to what non selective agonists are referring to? Beta 1, beta-2 agonists? In addition, no bibliographic source is given to support the statement.
Answer
Thank you for your comment. We have added the following information to support your comment ”This modulation is particularly relevant in cardiovascular diseases, in which the adrenergic pathway plays an hegemonic role in physiological cardiac function, mainly in β1 receptor (61), nevertheless, the exact receptor of tis pathway in which montanoa tomentosa acts it is still unknown, in contrast kaurane-type diterpenes, one of the multiple compounds of this genus, inhibit the contractility and induce relaxation of smooth muscle (60)”.
- The expression model with hypertrophy – should be corrected with cardiac hypertrophy.
Answer
We are changed this your suggestion in the manuscript.
Once again, I draw attention to the fact that the manuscript requires proofreading to correct typos, formatting, and grammatical errors.
- Autors state that: The exact biological mechanisms that underlie hypertension associated heart failure remain unclear, nevertheless, in patients with heart failure with preserved ejection fraction , it has been documented changes in proteins related to mitochondrial metabolism and the cardiac contractile apparatus, proteomics analysis of the left ventricular tissue showed an upregulation of keton bodies transporter, impairment in phosphorylation in proteins like titin, including changes in sarcomeric .........
This paragraph is supporting my previous request. First, the results do not come from patients with heart failure with preserved ejection fraction but from an mouse model of heart failure with preserved ejection fraction and the results are not correctly cited.
Answer
Thank you for your observation, We have added information to support this comment.